# Giant orbital magnetoelectric effect and current-induced magnetization switching in twisted bilayer graphene

Wen-Yu He [1]✉, David Goldhaber-Gordon[2,3] & K. T. Law [1]✉

Recently, quantum anomalous Hall effect with spontaneous ferromagnetism was observed in twisted bilayer graphenes (TBG) near 3/4 filling. Importantly, it was observed that an extremely small current can switch the direction of the magnetization. This offers the prospect of realizing low energy dissipation magnetic memories. However, the mechanism of the current-driven magnetization switching is poorly understood as the charge currents in graphenes are generally believed to be non-magnetic. In this work, we demonstrate that in TBG, the twisting and substrate induced symmetry breaking allow an out of plane orbital magnetization to be generated by a charge current. Moreover, the large Berry curvatures of the flat bands give the Bloch electrons large orbital magnetic moments so that a small current can generate a large orbital magnetization. We further demonstrate how the charge current can switch the magnetization of the ferromagnetic TBG near 3/4 filling as observed in the experiments.

[1] Department of Physics, Hong Kong University of Science and Technology, Clear Water Bay, Hong Kong, China. [2] Department of Physics, Stanford University, 382 Via Pueblo Mall, Stanford, CA 94305, USA. [3] Stanford Institute for Materials and Energy Sciences, SLAC National Accelerator Laboratory, 2575 Sand Hill Road, Menlo Park, CA 94025, USA. ✉email: wenyu@ust.hk; phlaw@ust.hk

A bilayer graphene with a twist angle $\theta$ between the two graphene layers forms a quasi-two-dimensional moiré superlattice, dramatically modifying its electronic properties[1-3]. At small twist angles $\theta$, the moiré potential effectively reduces the Dirac velocity[1,2] and yields flat bands at a series of magic angles[3], where electronic correlations become important. Recently, insulating[4] and superconducting phases[5], possibly driven by correlations at around 1/2 and −1/2 fillings in twisted bilayer graphenes (TBG) have been observed. These discoveries have stimulated intensive theoretical and experimental works[6-36] to understand the underlying insulating and superconducting mechanisms.

More recently, experiments have unveiled the topological properties brought about by the moiré potential, as the signatures of ferromagnetism and quantum anomalous Hall effect have been experimentally observed near the 3/4 filling[37,38]. These observations are consistent with predictions[7,24] that electron–electron interactions can give rise to ferromagnetism by lifting the spin and valley degeneracy, and that quantum anomalous Hall states will be obtained when bands with a total non-zero Chern number are filled. Strikingly, these experiments have also shown that the magnetization can be switched by driving very small DC currents (from 10 to 50 nA) through the samples[37,38]. The current needed for magnetization switching is several orders of magnitude smaller than those in state of the art spin-torque devices[39]. These observations strongly suggest the possibility of realizing ultralow power magnetic memory devices in TBG. However, it is not clear how a charge current can couple to the out-of-plane magnetization of the TBG, as the charge currents in graphene layers are generally believed to be non-magnetic.

Here, we show that charge currents in TBG can induce very large orbital magnetization at general filling factors even when the sample is not ferromagnetic. We call this effect the giant orbital magnetoelectric effect. First, by symmetry analysis, we point out that due to twisting, the symmetry of bilayer graphene is reduced from $D_{3d}$ (for AB bilayer graphene) or $D_{6h}$ (for AA stacking) to $D_6$ which belongs to the chiral point group. Thus, symmetry allows a magnetization to be induced by a charge current[40,41]. However, the $D_6$ symmetry of TBG is still too high to allow an out-of-plane magnetization to be generated by an in-plane current for current-induced magnetization switching. Importantly, we further note that closely aligning the hexagonal boron nitride (hBN) substrate to the TBG has been essential for experimental realization of ferromagnetism and the quantum anomalous Hall state[37,38]. Including substrate-induced sublattice symmetry breaking[24-27] and strain[29,34], the symmetry of the TBG is reduced to $C_1$ such that the applied current can induce a net out-of-plane magnetization[40]. Moreover, due to the large Berry curvature of the flat bands near the magic angle, the orbital magnetic moments carried by the Bloch electrons can be as large as tens of Bohr magnetons per electron even with very small strains. The lattice symmetry reduction and the large orbital magnetic moments of the electrons allow a large orbital magnetization to be induced by a small charge current. Near 3/4 filling when the Hall resistance $R_{xy}$ is not quantized and the longitudinal resistance $R_{xx}$ is finite (this is the experimental regime where current-induced magnetic switching has been observed[37,38]), the bulk conducting channels which carry magnetization can couple to the bulk magnetization of the sample, allowing current-controlled magnetic switching.

## Results

**Continuum model of strained TBG.** An isolated TBG can be described by coupling the top and bottom graphene layers with a twist angle $\theta$. Near the Fermi energy, the top and bottom graphene layers with the Dirac Hamiltonian at the valley $\xi$ can be described by a continuum model as[1-3]

$$\mathcal{H}_{t/b} = \hbar v_F \sum_{\mathbf{q},s,\xi} a^\dagger_{t/b,s,\xi}(\mathbf{q}) \hat{\mathbf{R}}_{\pm\frac{\theta}{2}} \mathbf{q} \cdot \boldsymbol{\sigma} a_{t/b,s,\xi}(\mathbf{q}), \tag{1}$$

where $\mathcal{H}_{t/b}$ denotes the Hamiltonian of the top and bottom layer respectively, $a^{(\dagger)}_{t/b,s,\xi}(\mathbf{q})$ is a two component creation (annihilation) operator creating (annihilating) electrons at the two A and B sublattices in the top/bottom graphene layer. The valley and the spin indices are denoted by $\xi = \pm 1$ and $s = \uparrow, \downarrow$ respectively. The momentum $\mathbf{q} = \mathbf{k} - \mathbf{K}_\xi$ is defined relative to the original Brillouin zone corner that hosts the Dirac point at $\mathbf{K}_\xi$, $d = 1.42\ \text{Å}$ is the carbon–carbon bond length[42], the rotation matrix has the form $\hat{\mathbf{R}}_{\pm\frac{\theta}{2}} = \cos\frac{\theta}{2} \mp i\sigma_y \sin\frac{\theta}{2}$, the Fermi velocity takes the value $\hbar v_F = 5.96\ \text{eV Å}$[42] and $\boldsymbol{\sigma} = (\sigma_x, \sigma_y)$ denotes the Pauli matrices.

An important effect of the moiré superlattice which originates from twisting is to fold the original Brillouin zone into the mini-Brillouin zones schematically shown in Fig. 1a. Both the $\mathbf{K}_+$ and $\mathbf{K}_-$ points of the original Brillouin zone are mapped to the mini-Brillouin zone, giving rise to fourfold degenerate minbands with both valley and spin degeneracy. In the reciprocal space, the Moiré superlattice has reciprocal vectors $\mathbf{q}_b = \frac{8\pi\sin\frac{\theta}{2}}{3\sqrt{3}d}(0,-1)$, $\mathbf{q}_{tr} = \frac{8\pi\sin\frac{\theta}{2}}{3\sqrt{3}d}\left(\frac{\sqrt{3}}{2}, \frac{1}{2}\right)$, $\mathbf{q}_{tl} = \frac{8\pi\sin\frac{\theta}{2}}{3\sqrt{3}d}\left(-\frac{\sqrt{3}}{2}, \frac{1}{2}\right)$ connecting the three neighboring sites of the hexagonal reciprocal lattice. The interlayer coupling is enabled when the momentum transfer between the Bloch states at different layers matches $\mathbf{q}_b$, $\mathbf{q}_{tr}$ or $\mathbf{q}_{tl}$. The interlayer coupling Hamiltonian of the continuum model[1-3] is present in the Methods section.

For an isolated TBG, the top and bottom graphene Hamiltonian along with the interlayer coupling respects the $D_6$ symmetry[7,11,16,18,19]. At the mini-Brillouin zone corner $\mathbf{K}^m_\pm$, two massless Dirac points emerge which are protected by the composite symmetry $C_2\mathcal{T}$ where $\mathcal{T}$ is the complex conjugate operator[7,11,18,19]. However, in the two recent experiments in which a ferromagnetic state has been seen, the TBG is coupled with a hBN cladding layer aligned to the TBG to less than 1°, which empirically appears necessary to support the ferromagnetism[37,38]. In our model, the hBN substrate affects the bottom graphene layer in two aspects: (1) it breaks the $C_2$ symmetry and introduces the massive gap $\Delta\sigma_z$ to the Dirac Hamiltonian as shown in Fig. 1b; (2) it exerts strain on the bottom graphene layer and further reduces the crystal symmetry to $C_1$.

In this work, for simplicity we use a uniaxial strain tensor $\mathcal{E}$ to describe the effect of strain. The strain tensor can be written as

$$\mathcal{E} = \varepsilon \begin{pmatrix} -\cos^2\phi + \nu\sin^2\phi & -(1+\nu)\cos\phi\sin\phi \\ -(1+\nu)\cos\phi\sin\phi & \nu\cos^2\phi - \sin^2\phi \end{pmatrix}, \tag{2}$$

with $\varepsilon$ being the tunable parameter to characterize the strain induced displacement, $\nu = 0.165$ the Poisson ratio for graphene[43], and $\phi$ the angle of the uniaxial strain relative to the zig-zag direction of the bottom graphene layer. In the presence of uniaxial strain, the real space and reciprocal space are transformed as $\tilde{\mathbf{r}} = (1 + \mathcal{E})\mathbf{r}$ and $\tilde{\mathbf{k}} = (1 - \mathcal{E}^T)\mathbf{k}$, respectively. Therefore the Dirac points in the bottom graphene layer are shifted from the original position $\mathbf{K}_\xi$ to $(1 - \mathcal{E}^T)\mathbf{K}_\xi - \xi\mathbf{A}$, where $\mathbf{A} = \frac{\beta}{d}\left(\mathcal{E}_{xx} - \mathcal{E}_{yy}, -2\mathcal{E}_{xy}\right)$ with $\beta = 1.57$ being the effective gauge field from the strain[44]. By combining the sublattice symmetry breaking and uniaxial strain effect from the hBN substrate, we are able to obtain the modified

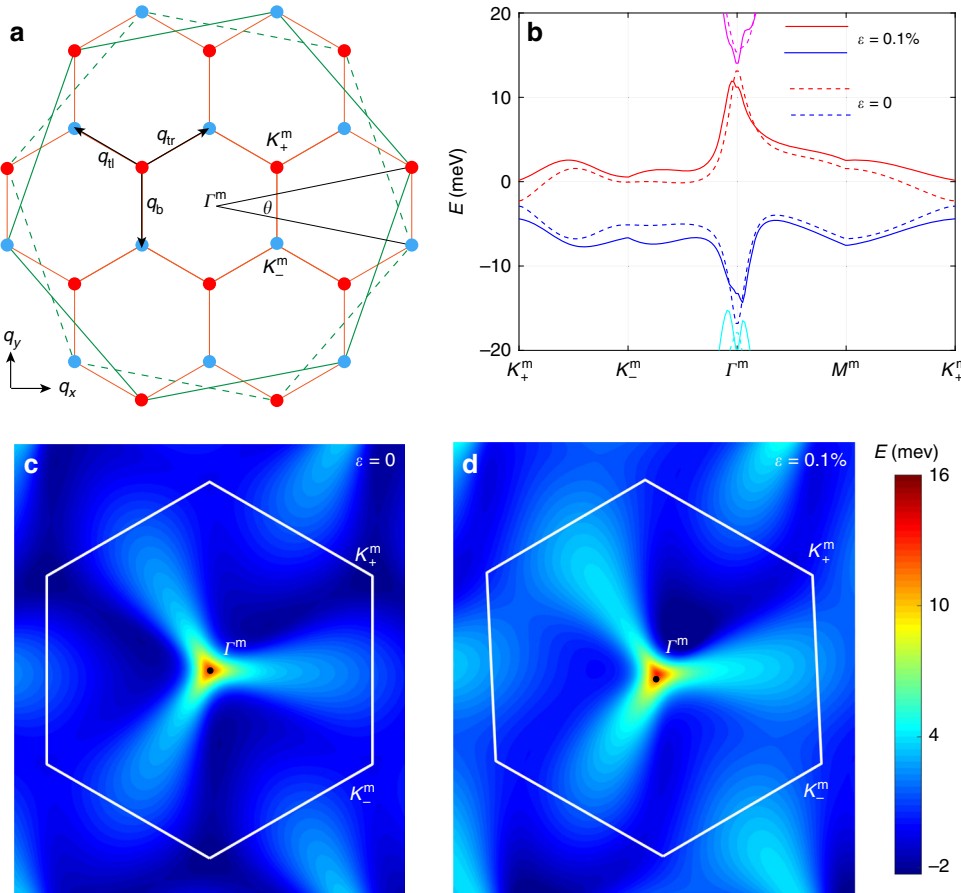

**Fig. 1 The energy dispersion of TBG. a** The original Brillouin zone is folded into the mini-Brillouin zone for the moiré superlattice. The solid and dashed green lines defining the large hexagons represent the original Brillouin zones of the top and bottom graphene layers respectively. **b** The flat bands energy dispersion from the $\mathbf{K}_+$ valley with the twist angle $\theta = 1.2°$. The dashed and solid lines are the cases with strain $\varepsilon = 0$ and $\varepsilon = 0.1\%$, respectively. The red and blue bands represent the conduction ($\nu = c$) and valence ($\nu = v$) bands, respectively. The conduction flat band energy dispersion in the mini-Brillouin zone with strain $\varepsilon = 0$ for **c** and $\varepsilon = 0.1\%$ for **d**, respectively. The energy dispersion from the other valley $\mathbf{K}_-$ can be mapped by the time-reversal symmetry as $E_{s,+,\nu}(\mathbf{q}) = E_{s,-,\nu}(-\mathbf{q})$.

bottom graphene layer Hamiltonian as

$$\tilde{\mathcal{H}}_{\mathrm{b}} = \sum_{\mathbf{q},s,\xi} a^{\dagger}_{\mathrm{b},s,\xi}(\mathbf{q}) \left[ \xi \hbar v_{\mathrm{F}} \hat{\mathbf{R}}_{-\frac{\theta}{2}} \left(1 + \mathcal{E}^{\mathrm{T}}\right)(\mathbf{q} + \xi \mathbf{A}) \cdot \boldsymbol{\sigma} + \Delta \sigma_z \right] a_{\mathrm{b},s,\xi}(\mathbf{q}),$$

(3)

where $\hat{\mathbf{R}}_{-\frac{\theta}{2}} = \cos\frac{\theta}{2} + i\sigma_y \sin\frac{\theta}{2}$ is the rotation matrix, the momentum $\mathbf{q} = \mathbf{k} - \left(1 - \mathcal{E}^{\mathrm{T}}\right)\mathbf{K}_\xi$ is defined relative to the uniaxial strain-deformed Brillouin zone corner, and the staggered potential introduced by the hBN substrate takes $\Delta = 17\,\mathrm{meV}$[45,46]. The staggered potential breaks the $C_{2z}$ symmetry and reduces $D_6$ to $D_3$ symmetry, while the uniaxial strain further removes all the crystal lattice symmetry and brings the TBG down to the $C_1$ group.

Denoting the interlayer coupling between the strained bottom layer graphene and the unstrained top layer graphene as $\tilde{\mathcal{H}}_{\mathrm{int}}$, the total Hamiltonian can be written as

$$\mathcal{H} = \tilde{\mathcal{H}}_{\mathrm{b}} + \mathcal{H}_{\mathrm{t}} + \tilde{\mathcal{H}}_{\mathrm{int}} = \sum_{\mathbf{q},s,\xi} A^{\dagger}_{s,\xi}(\mathbf{q}) h_\xi(\mathbf{q}) A_{s,\xi}(\mathbf{q}),$$

(4)

where $A_{s,\xi}(\mathbf{q})$ is a multicomponent operator and $h_\xi(\mathbf{q})$ is the Hamiltonian matrix as described in detail in the Methods section.

The energy dispersion at each band for the TBG can then be directly obtained through diagonalizing the continuum Hamiltonian in Eq. (4). For an isolated TBG, at the angle $\theta = 1.20°$, the Hamiltonian with $\mathcal{E} = 0$, $\Delta = 0$ gives the flat bands dispersion as shown in the dashed lines of Fig. 1b. The flat bands possess two

gapless Dirac points at $\mathbf{K}^m_\pm$ in the mini-Brillouin zone. The energy dispersion with strain is depicted by the solid lines in Fig. 1b. The energy dispersion for the conduction band (the red band in Fig. 1b) in the whole Brillouin zone are shown in Fig. 1c and d for the unstrained and the strained cases, respectively. It is clear from Fig. 1c that the energy dispersion $E_{s,\xi,\nu}(\mathbf{k})$ with spin index $s$, valley index $\xi$, and band index $\nu$ in general respects the $C_3$ symmetry as $E_{s,\xi,\nu}(\mathbf{q}) = E_{s,\xi,\nu}\left(\hat{\mathbf{R}}_{\frac{2\pi}{3}}\mathbf{q}\right)$. However, strain breaks the threefold rotational symmetry as shown in Fig. 1d, in which we set $\varepsilon = 0.1\%$ and the strain is applied along the direction of the zig-zag edge of the bottom layer graphene ($\phi = 0$ in Eq. (2)).

**Orbital magnetic moment in TBG.** With the Hamiltonian in Eq. (4), we can calculate the Berry curvature $\Omega^z_{s,\xi,\nu}(\mathbf{q})$ and the orbital magnetic moment in the out-of-plane direction $m^z_{s,\xi,\nu}(\mathbf{q})$ of each Bloch state:

$$\Omega^z_{s,\xi,\nu}(\mathbf{q}) = i\left\langle \partial_{\mathbf{q}} u_{s,\xi,\nu}(\mathbf{q}) \right| \times \left| \partial_{\mathbf{q}} u_{s,\xi,\nu}(\mathbf{q}) \right\rangle$$

(5)

and

$$m^z_{s,\xi,\nu}(\mathbf{q}) = \frac{ie}{2\hbar} \left\langle \partial_{\mathbf{q}} u_{s,\xi,\nu}(\mathbf{q}) \right| \times \left[ h_\xi(\mathbf{q}) - E_{\xi,\nu}(\mathbf{q}) \right] \left| \partial_{\mathbf{q}} u_{s,\xi,\nu}(\mathbf{q}) \right\rangle.$$

(6)

For an isolated TBG, the Berry curvature and the magnetic moment are non-zero only at the Dirac point. In the absence of

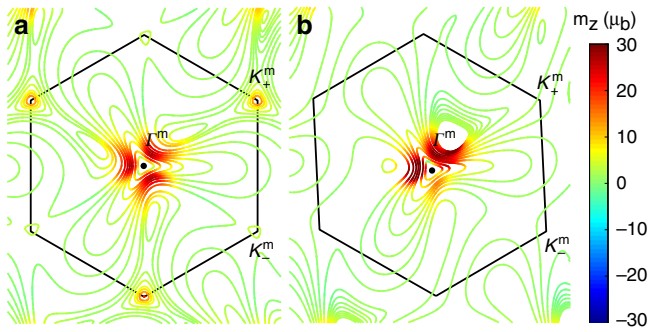

**Fig. 2 The orbital magnetic moments of the Bloch electrons. a** The orbital magnetic moments carried by the Bloch electrons in the mini-Brillouin zone with no strain. **b** The orbital magnetic moments of the electrons when a uniaxial strain characterized by $\varepsilon = 0.1\%$ is introduced. The staggered potential is set to be $\Delta = 17$ meV in both cases. In **b**, $C_3$ symmetry is broken and the Brillouin zone is deformed.

**Table 1 Magnetoelectric susceptibility pseudotensor $\alpha$ for $D_6$, $C_3$, and $C_1$ point group.**

| Point group | $\alpha$ | Point group | $\alpha$ |
|---|---|---|---|
| $D_6$ | $\begin{pmatrix} \alpha_\parallel & 0 & 0 \\ 0 & \alpha_\parallel & 0 \\ 0 & 0 & \alpha_{zz} \end{pmatrix}$ | $C_3$ | $\begin{pmatrix} \alpha_\parallel & -\alpha^- & 0 \\ \alpha^- & \alpha_\parallel & 0 \\ 0 & 0 & \alpha_{zz} \end{pmatrix}$ |
| $C_1$ | $\begin{pmatrix} \alpha_{xx} & \alpha_{xy} & \alpha_{xz} \\ \alpha_{yx} & \alpha_{yy} & \alpha_{yz} \\ \alpha_{zx} & \alpha_{zy} & \alpha_{zz} \end{pmatrix}$ | | |

$\alpha_{ij}$ with $i, j = x, y, z$ are in general the elements in $\alpha$. In $D_6$ and $C_3$, $\alpha_{xx} = \alpha_{yy}$ is denoted as $\alpha_\parallel = \alpha_{xx} = \alpha_{yy}$. In $C_3$, the antisymmetric off diagonal element is denoted as $\alpha^- = -\alpha_{xy} = \alpha_{yx}$.

strain but in the presence of the staggered potential which is set to be $\Delta = 17$ meV, the distribution of the orbital magnetic moment in the mini-Brillouin zone is shown in Fig. 2a. It respects the $C_3$ symmetry as $m^z_{s,\xi,\nu}(\mathbf{q}) = m^z_{s,\xi,\nu}\left(\hat{\mathbf{R}}_{\frac{2\pi}{3}}\mathbf{q}\right)$. The orbital magnetic moments are particularly large around $\mathbf{K}^m_\xi$ and $\mathbf{\Gamma}^m$ in the mini-Brillouin zone, where the flat band hybridizes with adjacent bands. The strength of the orbital magnetic moments can reach about $30\mu_b$ with $\mu_b = \frac{e\hbar}{2m_e}$ the Bohr magneton. In the presence of strain, the $C_{3z}$ symmetry of $m^z$ is broken as shown in Fig. 2b. If time-reversal symmetry is preserved, the orbital magnetic moment has the constraint $m^z_{s,+,\nu}(\mathbf{q}) = -m^z_{s,-,\nu}(-\mathbf{q})$, so that no net magnetization is allowed. However, due to the $C_3$ symmetry breaking, applying a current would create an imbalance in the magnetic moment distribution and thus a net out-of-plane magnetization[40] as demonstrated in the next section.

**Magnetoelectric response in TBG.** In quasi-two-dimensional materials with finite magnetoelectric response, the electric field induced magnetization can be described as

$$M_i = \sum_{i,j} \alpha_{ij} E_j, \tag{7}$$

with $i, j = x, y$, and $\alpha_{ij}$ the magnetoelectric susceptibility. As shown in refs. [40,41], the general forms of the components of the magnetoelectric susceptibility tensor $\alpha_{ij}$ can be determined by the crystal symmetry of the material. The general forms of $\alpha_{ij}$ for point groups $D_6$, $C_3$, and $C_1$ which are relevant to TBG are shown in Table 1. It is clear from Table 1 that it is possible to generate an out-of-plane magnetization by in-plane electric fields only if the crystal point group symmetry is reduced to $C_1$.

To calculate $\alpha_{ij}$ for TBG, we can use the linear response theory which gives[47,48]

$$\alpha_{ij} = -\tau \frac{e}{\hbar} \int_{\mathbf{q}} \sum_{s,\xi,\nu} M^i_{s,\xi,\nu}(\mathbf{q}) v^j_{s,\xi,\nu}(\mathbf{q}) f'(E_{s,\xi,\nu}), \tag{8}$$

where $\int_{\mathbf{q}} \equiv \frac{1}{(2\pi)^2} \int_{BZ} d\mathbf{q}$, $f(E)$ is the Fermi Dirac distribution function, $v^j_{s,\xi,\nu} = \partial_{q_j} E_{s,\xi,\nu}(\mathbf{q})$ is the group velocity, $\tau$ is the effective scattering time, and the total magnetic moment $\mathbf{M}_{s,\xi,\nu}(\mathbf{q}) = \mathbf{m}_{s,\xi,\nu}(\mathbf{q}) + \mathbf{S}_{s,\xi,\nu}(\mathbf{q})$ is composed of both the orbital magnetic moment $\mathbf{m}_{s,\xi,\nu}(\mathbf{q})$ and the spin magnetic moment $\mathbf{S}_{s,\xi,\nu} = \langle u_{s,\xi,\nu}(\mathbf{q})|\frac{1}{2}g\mu_b\boldsymbol{\sigma}|u_{s,\xi,\nu}(\mathbf{q})\rangle$ with the Lande g factor $g = 2$. To be specific, we apply a uniaxial strain with $\varepsilon = 0.1\%$ along the zig-zag edge direction of the bottom layer graphene. The

orbital magnetization in the Brillouin zone in the presence of strain is shown in Fig. 2b. The resultant magnetoelectric susceptibility can then be evaluated assuming the electron scattering time to be $\tau = 10$ ps[49]. For the conduction band $\nu = c$, the magnetoelectric susceptibility $\alpha_{zx}, \alpha_{zy}$ is shown in Fig. 3a as a function of the Fermi energy, where the Cartesian coordinate is set to have the $x$-axis along the angular bisector between the two zig-zag directions of the top and bottom graphene layers. The magnetoelectric susceptibility is maximized near the energy with the largest density of states. Interestingly, $\alpha_{zx}, \alpha_{zy}$ are still very large even when the density of states is very low. This is because the orbital magnetizations carried by the Bloch states near $\mathbf{\Gamma}^m$ are very large as a result of the Berry curvatures of the flat bands. This allows a large magnetization to be induced by a small current. As shown in the Supplementary Fig. 2, the current-induced orbital magnetization can be even stronger when strain is increased.

Assuming an external electric field of $10^4$ V/m, we obtain the out-of-plane magnetization under different electric field directions as shown in Fig. 3b, where the increasing radius in the polar plot denotes the Fermi energy increases from the conduction band bottom to the top. The magnetization can reach $0.02~\mu_b/\text{nm}^2$, 1–2 orders larger than in the largest Rashba spin–orbit coupling materials such as Au (111) surfaces and Bi/Ag bilayers[50,51]. The current-induced magnetization is anisotropic with respect to the direction of the current and it switches sign under reversal of the electric field. It is important to note that the current-induced magnetization discussed here can appear at a general filling factor even absent spontaneous ferromagnetism in the sample. This current-induced magnetization should be observable experimentally through optical Kerr effects as in the case of transition metal dichalcogenides[52].

**Current-induced magnetization switching in TBG.** TBG in the non-interacting limit possess valley and spin degeneracy for each flat band[1–3,7,11,18,19]. However, near the magic angles, the narrow band width at the Fermi level magnifies the role of interactions, and interaction-driven spontaneous symmetry breaking is observed experimentally[37,38]. Specifically, at 3/4 filling of the conduction band in hBN-aligned TBG with inter-graphene twist angle 1.20° (ref. [37]) a giant anomalous Hall effect of order $h/e^2$ has been reported; and for TBG with twist angle 1.15°[38], quantized anomalous Hall effect has been reported, in both cases at zero external magnetic field. Hysteresis in the Hall conductance under out-of-plane magnetic fields suggests spontaneous ferromagnetism with out-of-plane magnetization.

The presence of net magnetization as revealed by anomalous Hall resistance[37,38] indicates that the spin and/or valley degeneracies are lifted, possibly by interactions[7,24–27]. As a result, there are four bands (which originated from the fourfold

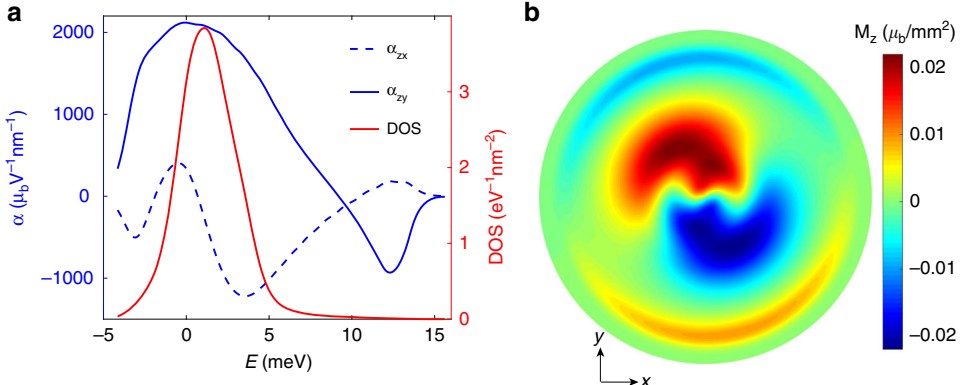

**Fig. 3 The magnetoelectric response in strained TBG. a** The magnetoelectric susceptibilities $\alpha_{zx}$, $\alpha_{zy}$, and the density of states (DOS), both as a function of the Fermi energy $E_F$ from bottom to top of the conduction band. We have set $\varepsilon = 0.1\%$. **b** The induced magnetization at the electric field strength $10^4$ V/m along different in-plane angles. Increasing radius in the polar plot denotes the Fermi energy increasing from the conduction band bottom to the top. The twist angle is set to be $\theta = 1.2°$.

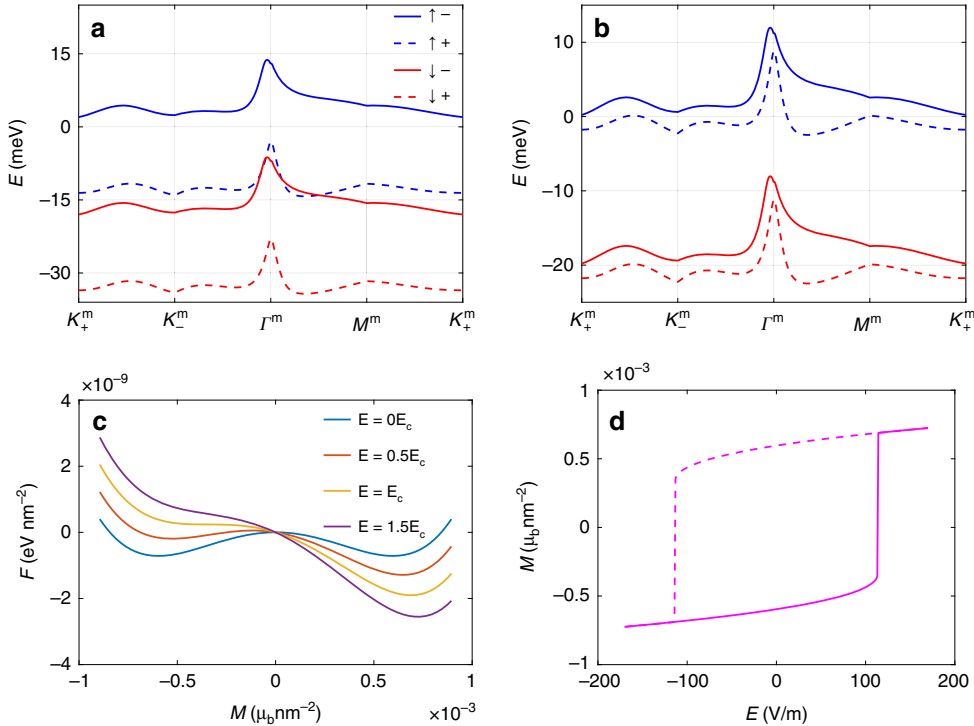

**Fig. 4 Current induced magnetization switching.** The interaction-renormalized bands at 3/4 filling for heterostrain $\epsilon = 0.1\%$ are shown in **a** for fully-filled spin- and valley-polarized bands, and in **b** for spin-polarized but valley-unpolarized bands. **c** Free energy as a function of magnetization for several values of applied electric field along the $y$ direction. At the coercive electric field $E_c$, one local minimum of the free energy collapses and only one minimum remains. **d** The magnetic hysteresis curve induced by the electric field along the $y$ direction. The coercive field is estimated to be around $E_c = 113$ V/m.

degenerate conduction band in Fig. 1b) labeled by the spin indices $s = \uparrow, \downarrow$ and valley indices $\xi = \pm 1$ available for the electrons to fill as shown in Fig. 4a and b. To take into account the simplest possible spin and valley polarization phenomenologically, the dispersion of the four bands is written as $E_{s,\xi,c}(\mathbf{q}) = E_c(\xi\mathbf{q}) - \mu_{s,\xi}$ where $E_c(\xi\mathbf{q})$ describes the original conduction band dispersion from valley $\xi$ without interaction and $\mu_{s,\xi}$ is the spin- and valley-dependent energy shift due to interactions.

At filling factor 3/4, if the three bands with lower energy are completely filled as depicted in Fig. 4a, the TBG should display the quantum anomalous Hall effect. At the same filling factor 3/4, the top two bands could instead each be partially filled as seen in Fig. 4b, in which case the TBG would have a bulk conducting channel in parallel with the anomalous Hall conductance. The scenario of Fig. 4b may be a good representation of experiments where the Hall conductance is not quantized and bulk conducting channels exist[37].

To connect our theory with experiments, we note that the spontaneous ferromagnetism in TBG can be described by the Landau's free energy density as

$$F = -a_0 M_z^2 + b_0 M_z^4 - M_z B_z. \tag{9}$$

Below the critical temperature, $a_0 > 0$, $b_0 > 0$, generating a finite magnetization order parameter $M_z = \sqrt{\frac{a_0}{2b_0}}$ at $\mathbf{B} = 0$ and the magnetic susceptibility reads $\chi_{zz} = \frac{1}{4a_0}$. In the presence of external

magnetic field $B_z$, the magnetization switches sign at the coercive magnetic field $B_c = \frac{4a_0}{3}\sqrt{\frac{a_0}{6b_0}}$. Note that $a_0$ and $b_0$ can be obtained once $M_z$ and $\chi_{zz}$ are calculated using the continuum model introduced previously with the energy of the bands shifted by $\mu_{s,\xi}$. Given $\mu_{s,\xi}$, the total magnetization $M_z$ and the magnetic susceptibility $\chi_{zz}$ can be evaluated as

$$M_z = \int_{\mathbf{q}} \sum_{s,\xi} M^z_{s,\xi,c}(\mathbf{q}) f\left[E_c(\xi\mathbf{q}) - \mu_{s,\xi}\right], \quad (10)$$

$$\chi_{zz} = -\int_{\mathbf{q}} \sum_{s,\xi} \left[M^z_{s,\xi,c}(\mathbf{q})\right]^2 f'\left[E_c(\xi\mathbf{q}) - \mu_{s,\xi}\right], \quad (11)$$

where $M^z_{s,\xi,c}$ is the z-component of the total magnetic moment of a Bloch wavefunction of the flat bands. In the partially polarized state shown in Fig. 4b with $\{\mu_{\uparrow,-}, \mu_{\uparrow,+}, \mu_{\downarrow,-}, \mu_{\downarrow,+}\} = \{-0.01, 2.4, 20, 22.4\}$ meV, we find that $a_0 = 4 \times 10^{-3}\ \mu_b^{-2}$ eV nm$^2$, $b_0 = 5.65 \times 10^3\ \mu_b^{-4}$ eV nm$^6$ and the coercive magnetic field $B_c = 31.8$ mT.

To understand the coupling between the electric field and the magnetic field, we note that the total magnetization $M_z$ is changed to $M_z + \delta M_z$ where $\delta M_z = \alpha_{zx}E_x + \alpha_{zy}E_y$ is the magnetization induced by the current. As a result, the Landau free energy in the presence of an electric field can be written as

$$\begin{aligned}F &= -a_0(M_z + \delta M_z)^2 + b_0(M_z + \delta M_z)^4 \\ &\approx -a_0 M_z^2 + b_0 M_z^4 - 2a_0 M_z(\alpha_{zx}E_x + \alpha_{zy}E_y),\end{aligned} \quad (12)$$

which clearly shows that the magnetization of the sample couples to the electric field. Figure 4c depicts the free energy landscape as a function of magnetization changes for different electric field strength, using realistic parameters. By assuming the current is passed in the y-direction and by calculating $\alpha_{zy}$, the resulting hysteresis loop of magnetization as a function of electric field is determined. The minimal electric field needed to switch the magnetization is estimated to be about 113 V/m. In a recent experiment[37], the longitudinal resistance is measured to be $R_{xx} = 10$ kΩ and the length between the contacting leads is estimated to be 5 μm. As a result, the coercive electric field at $E_c = 113$ V/m gives the coercive DC current $I_c = 57$ nA, which matches well with the experimental values of $30 - 40$ nA[37]. Since many of the details such as the strain, the band structure of the sample, the shifts of the polarized bands, etc. will affect the coercive current, the specific value of the coercive electric field calculated here can only be a rough estimation.

## Discussion

In the above sections, using a continuum model of TBG and incorporating the effects of sublattice symmetry breaking and strain, the magnetoelectric response was calculated. Here, we would like to emphasize that the analysis based on symmetry is very general. The exact form of the strain is not important. The breaking of the $D_6$ symmetry can come from other sources such as spatial inhomogeneity in the chemical potential or twist angles. The detailed source of symmetry breaking will not affect our conclusion that currents can induce magnetization in TBG. Moreover, the current-induced magnetization effect can appear even when the system itself is not ferromagnetic (for example, in the absence of valley polarization). Therefore, we expect that other materials with low crystal symmetries such as twisted bilayer-bilayer graphene[53–55], twisted hBN-graphene heterostructure[56,57], twisted transition metal dichacolgenides[58], and gapped bilayer graphene[59] with strain will exhibit similar magnetoelectric effects, although the magnitude of the

magnetoelectric response will depend on the details of the materials. The current-induced orbital magnetization predicted can be tested by magneto-optical Kerr effect in experiments[52].

Another important point is that in the experimental regime where current-induced magnetization switching is demonstrated, the Hall resistance $R_{xy}$ is not quantized and the longitudinal resistance $R_{xx}$ is finite[37,38]. The currents can flow between domains with different magnetization. As the symmetry of the problem is still $C_1$ even including the domains, the bulk currents can carry out-of-plane magnetization and switch the magnetization of the domains. However, a calculation incorporating domains is beyond the scope of the current study.

Our picture of current-induced magnetic switching does not apply directly to quantum anomalous Hall states with an insulating bulk when the current is carried by the edge states only. To obtain the current-induced magnetization, we assumed that the scattering time ($\tau$) in the system is finite as shown in Eq. (8). This assumption does not apply to chiral edge states. Serlin et al.[38] argued that even edge states which do not carry net magnetization can also switch the direction of the magnetic domains, giving an effect proportional to $I^3$ where $I$ is the current carried by the edge states. In contrast, in the present work, the magnetoelectric effect of the bulk currents couples the electric field linearly to the magnetization as shown in Eq. (12).

It is also worth noting that the current-induced magnetization in TBG is purely orbital in nature. It is different from the magnetoelectric effect induced by spin–orbit coupling in non-centrosymmetric materials[60,61] studied previously. It is also interesting to note that the orbital magnetization can be strongly affected by strain. In this work, we only discussed the strain induced naturally by the hBN substrate. Experimentally, one can induce a much larger strain on the TBG artificially. In this case, the current-induced magnetization could be further enhanced. The orbital magnetization of some of the Bloch states in the Brillouin zone can even reach a hundred Bohr magnetons with moderate strain as shown in the Supplementary Fig. 1. In this case, even larger orbital magnetoelectric effects could be realized in TBG.

## Methods

**Interlayer coupling Hamiltonian for the TBG.** In the continuum model description[1–3], the state at $\mathbf{q}$ from one layer will couple with the state at $\mathbf{q}'$ from the other layer if $\mathbf{q} - \mathbf{q}'$ matches $\mathbf{q}_b$, $\mathbf{q}_{tr}$, or $\mathbf{q}_{tl}$, so the interlayer coupling Hamiltonian reads

$$\mathcal{H}_{int} = \sum_{\mathbf{q},\mathbf{q}',s,\xi} a^\dagger_{t,s,\xi}(\mathbf{q})\left[T_{\xi\mathbf{q}_b}\delta_{\mathbf{q}'-\mathbf{q},\xi\mathbf{q}_b} + T_{\xi\mathbf{q}_{tr}}\delta_{\mathbf{q}'-\mathbf{q},\xi\mathbf{q}_{tr}} + T_{\xi\mathbf{q}_{tl}}\delta_{\mathbf{q}'-\mathbf{q},\xi\mathbf{q}_{tl}}\right]a_{b,s,\xi}(\mathbf{q}') + \text{h.c.}, \quad (13)$$

with the tunneling matrix

$$T_{\mathbf{q}_b} = \frac{1}{3}t_\perp\begin{pmatrix} 1 & 1 \\ 1 & 1 \end{pmatrix}, \quad (14)$$

$$T_{\mathbf{q}_{tr}} = \frac{1}{3}t_\perp\begin{pmatrix} 1 & e^{\frac{i2\pi}{3}} \\ e^{-\frac{i2\pi}{3}} & 1 \end{pmatrix}, \quad (15)$$

$$T_{\mathbf{q}_{tl}} = \frac{1}{3}t_\perp\begin{pmatrix} 1 & e^{-\frac{i2\pi}{3}} \\ e^{\frac{i2\pi}{3}} & 1 \end{pmatrix}. \quad (16)$$

In the presence of uniaxial strain $\mathcal{E}$ in the bottom layer graphene, the reciprocal vectors are deformed as $\mathbf{q}_b \to \tilde{\mathbf{q}}_b$, $\mathbf{q}_{tr} \to \tilde{\mathbf{q}}_{tr}$, $\mathbf{q}_{tl} \to \tilde{\mathbf{q}}_{tl}$ and the tunneling matrix are modified as $T_{\mathbf{q}_b} \to \tilde{T}_{\tilde{\mathbf{q}}_b}$, $T_{\mathbf{q}_{tr}} \to \tilde{T}_{\tilde{\mathbf{q}}_{tr}}$, $T_{\mathbf{q}_{tl}} \to \tilde{T}_{\tilde{\mathbf{q}}_{tl}}$, where the detailed forms are presented in the Supplementary Note 1.

**The Hamiltonian matrix for TBG coupled with hBN substrate.** The Hamiltonian for the TBG on a hBN substrate reads

$$\mathcal{H} = \sum_{\mathbf{q},s,\xi} A^\dagger_{s,\xi}(\mathbf{q})h_\xi(\mathbf{q})A_{s,\xi}(\mathbf{q}), \quad (17)$$

where $A_{s,\xi}(\mathbf{q})$ has infinite components representing the series of states $a_{\mathrm{b},s,\xi}(\mathbf{q})$, $a_{\mathrm{t},s,\xi}(\mathbf{q}')$ with $\mathbf{q} - \mathbf{q}' = \xi\mathbf{q}_{\mathrm{b}}, \xi\mathbf{q}_{\mathrm{tr}}, \xi\mathbf{q}_{\mathrm{tl}}$. For example, in the truncated basis of $[a_{\mathrm{b},s,\xi}(\mathbf{q}), a_{\mathrm{t},s,\xi}(\mathbf{q} + \xi\mathbf{q}_{\mathrm{b}}), a_{\mathrm{t},s,\xi}(\mathbf{q} + \xi\mathbf{q}_{\mathrm{tr}}), a_{\mathrm{t},s,\xi}(\mathbf{q} + \xi\mathbf{q}_{\mathrm{tl}})]^{\mathrm{T}}$, the Hamiltonian matrix $h_{\xi}(\mathbf{q})$ has the form:

$$h_{\xi}(\mathbf{q}) = \begin{pmatrix} h_{\mathrm{b},\xi}(\mathbf{q}) & \tilde{T}_{\xi\tilde{\mathbf{q}}_{\mathrm{b}}} & \tilde{T}_{\xi\tilde{\mathbf{q}}_{\mathrm{tr}}} & \tilde{T}_{\xi\tilde{\mathbf{q}}_{\mathrm{tl}}} \\ \tilde{T}^{\dagger}_{\xi\tilde{\mathbf{q}}_{\mathrm{b}}} & h_{\mathrm{t},\xi}(\mathbf{q} + \xi\tilde{\mathbf{q}}_{\mathrm{b}}) & 0 & 0 \\ \tilde{T}^{\dagger}_{\xi\tilde{\mathbf{q}}_{\mathrm{tr}}} & 0 & h_{\mathrm{t},\xi}(\mathbf{q} + \xi\tilde{\mathbf{q}}_{\mathrm{tr}}) & 0 \\ \tilde{T}^{\dagger}_{\xi\tilde{\mathbf{q}}_{\mathrm{tl}}} & 0 & 0 & h_{\mathrm{t},\xi}(\mathbf{q} + \xi\tilde{\mathbf{q}}_{\mathrm{tl}}) \end{pmatrix}, \quad (18)$$

with

$$h_{\mathrm{t},\xi}(\mathbf{q}) = \xi\hbar v_{\mathrm{F}}\hat{R}_{\frac{\theta}{2}}\mathbf{q} \cdot \boldsymbol{\sigma}, \quad (19)$$

$$h_{\mathrm{b},\xi}(\mathbf{q}) = \xi\hbar v_{\mathrm{F}}\hat{R}_{-\frac{\theta}{2}}(1 + \mathcal{E}^{\mathrm{T}})(\mathbf{q} + \xi\mathbf{A}) \cdot \boldsymbol{\sigma} + \Delta\sigma_z. \quad (20)$$

In our calculation, we consider 42 sites in the hexagonal reciprocal lattice and $h_{\xi}(\mathbf{q})$ is a $84 \times 84$ matrix in the truncated basis. The Hamiltonian matrix for the TBG respects the time-reversal symmetry as

$$h_{+}(\mathbf{q}) = h_{-}^{*}(-\mathbf{q}), \quad E_{s,+,\nu}(\mathbf{q}) = E_{s,-,\nu}(-\mathbf{q}), \quad (21)$$

with $E_{s,\xi,\nu}(\mathbf{q})$ the energy dispersion for the band index $\nu$, valley index $\xi$, spin index $s$.

## Data availability
The data that support the findings of this study are available from the corresponding author upon reasonable request.

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

## Acknowledgements

The authors thank Leon Balents, Xi Dai, Pablo Jarillo-Herrero, Patrick Lee, Kin Fai Mak, Senthil Todadri, and Andrea Young for the inspiring discussions. W.-Y.H. and K.T.L. are thankful for the support of HKRGC through C6026-16W, 16307117 and 16309718, 16310219. K.T.L. is further supported by the Croucher Foundation and the Dr. Tai-chin Lo Foundation. D.G.-G.'s work was supported by the U.S. Department of Energy, Office of Science, Basic Energy Sciences, Materials Sciences and Engineering Division, under Contract No. DE-AC02-76SF00515.

## Author contributions

K.T.L. and D.G.-G. conceived the idea and initiated the project. W.-Y.H. performed the theoretical calculations. All the authors discussed the results and co-wrote the paper.

## Competing interests

The authors declare no competing interests.
