## [Peer Review File · Nature Communications]

Reviewers' comments:

Reviewer #1 (Remarks to the Author):

There has been plenty of activity and excitement in the field of twisted graphene bilayers since the experimental observation of correlated insulating phases combined with superconductivity. This manuscript is devoted to another recent exciting experimental development, namely indication of ferromagnetism and quantum anomalous Hall effect at $3/4$ filling, and evidence that application of small currents induce a reversible magnetization in the sample.

The main claim of the paper is that the lattice symmetry reduction in twisted graphene bilayers by itself promotes a large out-of-plane magnetization per valley. When the valley degeneracy is lifted presumably by interaction effects (spontaneous broken symmetry), a large net magnetization can emerge. Those effects are claimed to be further enhanced by strain and sublattice symmetry breaking. The model is based on a continuum calculation of twisted graphene bilayers proposed in ref. 3-5. Substrate effects are accounted by breaking the C_2^{\prime} symmetry, which protects the Dirac points of the moire BZ, resulting in the opening of a gap, and also with strain effects, which displace those points and break C_3 symmetry.

The understanding that gapped graphene bilayers in general can have a giant magnetoelectric effect when the valley degeneracy is explicitly broken (for instance in the quantum anomalous Hall state or else by selectively exciting valleys with light in dichalcogenides) has been known for a while and I would say is hardly surprising by now (see for instance Nandkishore et al. PRL 107,097402 (2011)). The perhaps more interesting and nontrivial question, which is not addressed in this paper, is why the putative quantum anomalous Hall phase emerges in the first place.

With that in mind, my assessment is that the calculation of the magnetization through the Berry curvature is interesting and timely, but not sufficiently original in my view to justify publication in Nature Communications. I would rather recommend publication in a more specialized journal.

Reviewer #2 (Remarks to the Author):

In the manuscript 'Giant Orbital Magnetoelectric effect and Current-induced Magnetization Switching in Twisted Bilayer Graphene', the authors studied how the out-of-plane magnetic moment could switch under the in-plane electrical current. The authors constructed a model for the band structure in the presence of strain and the underlying substrate of BN. The authors found that the out-of-plane orbital magnetic moment of the Bloch states could become huge when the momentum is close to Γ . Motivated by the recent experiments on magnetic moment switching by electrical current, the authors also calculated the magnetoelectrical response based on the linear response theory. The authors also proposed a phenomenological model for current switching. I recommend its publication on nature communication as this article contains some impressive points:

1. The authors notice that the system is not in the quantum anomalous Hall regime when the current switching occurs. Although it is generally believed that the insulating state at commensurate filling could be understood from strong coupling limit, This observation partially justifies the authors' approach that treats the system as Fermi liquid. Furthermore, this model leads to predictions different from the theory based on edge modes.

2. This paper also contains some interesting points on symmetry analysis and reveals that the symmetry should be largely broken in order to explain the observed magnetoelectrical effects. This

symmetry broken could result from the strain and aligned BN. The authors constructed a nice model for the strain effects.

Overall, it is a nice and interesting article. Therefore, I do recommend its publication on nature communication.

Summary of Reply:

We thank both referees for their careful reading of the manuscript. We are gratified by Referee 2's appreciation of our work, and his/her enthusiastic support for publication in Nature Communications.

Referee 1 made many useful comments, which helped us realize where we had not clearly explained some subtle but important points. Based on this invaluable input, we have now improved our explanations. However, we believe that the key novel findings of our manuscript have been overlooked by Referee 1. Specifically, the giant magnetoelectric effect (current-induced magnetization effect) we studied in twisted bilayer graphene (TBG) is *fundamentally different* from the large magnetization effect caused by valley imbalance in gapped bilayer graphene.

Moreover, we would like to emphasize that, the current-induced magnetic switching in TBG is a very interesting and important phenomenon. The current needed to switch a magnetic bit of a ferromagnetic TBG is in the order of 10nA. This is six orders of magnitude smaller than the current needed to switch the magnetization direction in state of the art spin-orbit torque devices, which require currents of about 10mA for magnetic switching [Nat. Nanotech. 11 878(2016)]. This phenomenon has great potential applications in current-controlled magnetic memory devices and it deserves a clear theoretical explanation.

In our work, we have not only explained the mechanism of the current-induced switching in TBG, but also predicted that many other moire systems can also have large current-induced magnetization. We believe that our work provides a clear theoretical guidance for the study of the current-induced magnetization effects in moire materials.

In our reply to the referees below, we also highlight the difference between our work and the work by Nandkishore and Levitov mentioned by Referee 1. We hope that our clarification here and in more detail below helps Referee 1 reassess the importance of our work.

Reply to the Comments of the Referees.

Referee 1:

Comment 1: “The main claim of the paper is that that the lattice symmetry reduction in twisted graphene bilayers by itself promotes a large out-of-plane magnetization per valley. When the valley degeneracy is lifted presumably by interaction effects (spontaneous broken symmetry), a large net magnetization can emerge. Those effects are claimed to be further enhanced by strain and sub-lattice symmetry breaking.”

Reply 1: We thank the Referee for the careful reading of the manuscript.

The Referee points out correctly that each valley of twisted bilayer graphene (TBG) can carry a large out-of-plane magnetization (implicitly recognizing that the magnetization is opposite for each valley so that the net magnetization is zero). The Referee then correctly notes that “when the valley degeneracy is lifted presumably by interaction effects (spontaneous broken symmetry), a large net magnetization can emerge”.

We differ however with the Referee on the role these observations play in our manuscript: The main intent of our work is *not* to show that valley polarization in TBG will cause net magnetization. As the Referee points out [Nandkishore et al. PRL 107,097402 (2011)], this connection is already well-known. Nor is the claim that lattice symmetry reduction can enhance the valley polarization central to our narrative.

Instead, the main contribution of our work is to provide an explanation for why an exceedingly small current (6 orders of magnitude smaller current than that needed for switching a magnetic bit using state of the art spin-orbit torque) is capable of switching the magnetization direction of TBG when the TBG is ferromagnetic. This behavior was observed in two recent experiments (by Goldhaber Gordon's group at Stanford [Science 365, 605-608 (2019)] and Young's group at UCSB [Science.aay5533]). These striking experimental findings were not understood nor explained theoretically previously. In the following, we would like to show the importance and broad interest of our work:

i) Explaining current-induced magnetization switching in TBG:

To explain the current-induced magnetization switching observed, there are three important steps laid out in our manuscript (we have revised the manuscript to try to make these steps clearer and more prominent.)

First, even *without* valley polarization and with the TBG being non-ferromagnetic, a small current can induce a large out-of-plane magnetization in TBG (this is very different from the case of conventional gapped bilayer graphene, where the current-induced magnetization is zero).

Second, this large current-induced magnetization can happen in TBG when:

- a. The three-fold rotational symmetry in TBG is broken (by strain, for example).
- b. The orbital magnetization of the Bloch electrons is large (due to the large Berry phase of the flat bands of TBG). And the strain further enhances the orbital magnetization.
- c. The density of states of the flat band is large at the Fermi energy.

We pointed out through our calculations that these three conditions can be satisfied by TBG coupled to boron nitride substrates. It is important to note that this large current-induced magnetization effect can happen at a general filling factor of the TBG, whether or not the TBG is in a quantum anomalous Hall phase. Moreover, the direction of the magnetization is reversed when the direction of the current is reversed, just as seen in experiment when magnetization can be detected through anomalous Hall measurements.

Third, when there is ferromagnetism in TBG (such as near the $\frac{3}{4}$ filling), the magnetization induced by current can couple to the magnetization of the TBG and result in current-controlled magnetization switching. Therefore, our theory accounts for the experimental findings of Goldhaber-Gordon and Young regarding current-induced switching.

ii) The novelty and importance of our work.

1. Our work explains why an extremely small current can control the magnetization direction of a ferromagnetic TBG which is an important experimental finding with possible applications in ultra-low dissipation magnetic memory devices.

With our understanding, we also predicted that there is current-induced magnetization effect in other moire materials such as in twisted transition metal dichalcogenides (TMDs), monolayer graphene coupled to boron nitride substrate etc. This work will lead experimentalists and theorists to further study the current-induced magnetization in moire materials.

2. The current-induced magnetization effect discussed in our work is a very special kind of magnetoelectric effect.

As discussed at the end of the Discussion Section, current-induced magnetization has been studied in non-centrosymmetric materials with spin-orbital coupling three decades ago [1]. This effect is also called the Edelstein effect. The magnetization induced is due to spin magnetic moments. However, the current-induced magnetization is zero when the spin-orbit coupling is zero. In our work, we showed that in TBG, the current-induced magnetization is very large even in the absence of spin-orbit coupling. And the magnetization induced is purely orbital in nature. This kind of orbital magnetization was missed in Edelstein's theory which ignored the effect of Berry curvature of the band. In TBG, the large orbital Edelstein effect is caused by the large Berry curvature and also the large density of states of the flat bands.

3. In the Reply to Comment 3, we will show that the current-induced magnetization in TBG is very different from the magneto-optical Kerr effect studied by Nandkishore and Levitov for gapped bilayer graphene.

Comment 2: "The model is based on a continuum calculation of twisted graphene bilayers proposed in ref. 3-5. Substrate effects are accounted by breaking the C_2^{\prime} symmetry, which protects the Dirac points of the moire BZ, resulting in the opening of a gap, and also with strain effects, which displace those points and break C_3 symmetry."

Reply 2: We thank the Referee for a nice summary of our calculations.

Comment 3: "The understanding that gapped graphene bilayers in general can have a giant magnetoelectric effect when the valley degeneracy is explicitly broken (for instance in the quantum anomalous Hall state or else by selectively exciting valleys with light in dichalcogenides) has been known for a while and I would say is hardly surprising by now (see for instance Nandkishore et al. PRL 107,097402 (2011))."

Reply 3: We agree with the Referee that gapped graphene bilayers in general can have a large magnetization when the valley degeneracy is explicitly broken. But this is not the giant magnetoelectric effect that we studied in the present work.

The giant magnetoelectric effect in our manuscript refers instead to the large orbital magnetization induced by a small current. In the case of gapped graphene bilayers discussed by Nandkishore and Levitov [PRL 107,097402 (2011)], the current-induced

magnetization effect is indeed zero due to the C_3 rotational symmetry. If the C_3 rotational symmetry in a conventional gapped graphene bilayer is broken by strain, the induced magnetization at a given current is three orders of magnitude smaller than the case of strained TBG that we studied (as shown in Fig.R1 below). In the following, we provide a detailed account of why TBG is very different from gapped bilayer graphene.

i) The difference our work reveals between TBG and the conventional gapped bilayer graphene mentioned by the Referee.

First, in the case of conventional gapped bilayer graphene without net valley polarization, no matter how large the orbital magnetizations of the Bloch electrons are, a charge current cannot induce an out-of-plane magnetization. The C_3 symmetry will enforce the out-of-plane magnetization to be exactly zero. This is also the case for conventional TMDs.

Second, in the case of conventional gapped bilayer graphene with valley polarization, the current can carry magnetization due to the valley polarization. However, the magnetization direction of the current will always be the same as the magnetization of the sample, so the charged current cannot switch the magnetization direction of the sample.

Third, when strain is applied to conventional gapped bilayer graphene to break the C_3 rotational symmetry, symmetry allows finite current-induced magnetization as in the case of TBG (to the best of our knowledge, this problem has not been studied previously). However, as we show in the following figure (Fig.R1), the current-induced magnetization in this case is three orders of magnitude smaller than the twisted bilayer graphene case. This is due to the small density of states of gapped bilayer graphene. Though the orbital magnetic moments of the Bloch electrons near the band bottom of gapped bilayer graphene can be as large as 15 Bohr magneton near K and $-K$ points (Fig.R1b), these moments decrease quickly away from these high-symmetry points. Near the K points, the only places where the moments are large, the density of states is very small. Therefore, the current-induced magnetization is very small. In Fig.R1c, we show that with 0.1% strain, the current-induced magnetization is three orders of magnitude smaller than the case of TBG.

Fig.R1 a) The band structure of gapped bilayer graphene. The gap size is set to be 200 meV as shown in a recent experiment [2, 3]. b) The orbital magnetic moment of the Bloch electrons of gapped bilayer graphene. As pointed out by the referee, the orbital magnetic moments can be as large as 15 Bohr magneton near the conduction band bottom. However, the magnetic moment decreases quickly away from the K points. c) The magneto-electric response of a gapped bilayer graphene. When the strain is zero

(red line), the current-induced magnetization is always zero. Even when a strain is applied (blue curve), the current-induced magnetization is three orders of magnitude smaller than the case of TBG: The tensor element α , which characterizes the strength of the current-induced magnetization as defined in Eq.7 of the main text, can be as large as 2000 in strained TBG (Fig.3a of the main text) as compared to about 2.5 in gapped bilayer graphene. In the calculation for c), we considered a current direction which maximizes the magnetization. One can also show that α is not sensitive to the gap size which is set to be 200meV in the calculations.

ii) The difference between our work and the work by Nandkishore and Levitov.

Since the Referee specifically referenced the work by Nandkishore and Levitov as anticipating our present work, we would like to highlight the strong differences:

Nandkishore et al. pointed out that:

1. If a bilayer graphene is in the quantum anomalous Hall state, that bilayer graphene can induce finite magneto optical Kerr rotation.
2. In the nematic phase of a bilayer graphene, due to the breaking of the C_3 rotational symmetry, the intensity of the reflected light breaks the C_3 symmetry as well.

The results of the elegant work by Nandkishore et al. are useful for optical detections of spontaneous symmetry breaking phases of suspended bilayer graphene, for which transport measurements can be difficult to perform. Their results are related to the photon-induced optical transition between the bands of the gapped bilayer graphene. In our work, the current-induced magnetization is related instead to the properties of the Fermi surface. The work by Nandkishore et al. is not directly relevant to the current induced magnetization in gapped bilayer graphene. Their results cannot be used to predict the current-induced magnetization and magnetization switching in TBG.

Comment 4: “The perhaps more interesting and nontrivial question, which is not addressed in this paper, is why the putative quantum anomalous Hall phase emerges in the first place.”

Reply 4: We agree with the Referee that understanding why the quantum anomalous Hall phase emerges in the first place is an extremely interesting question. Indeed, preliminary results have been obtained by a few authors (see Refs. 9, 26-29 in the manuscript).

However, the understanding of the current-induced magnetization switching in TBG, which makes TBG a potential platform for ultra-low energy dissipation magnetic memory devices, is equally important both for fundamental science and for device applications. As pointed out in both Goldhaber-Gordon and Young’s works, the magnitude of current needed to switch a magnetic bit is orders of magnitude smaller than any other spin-orbital torque devices. This is an extremely important experimental observation which requires a theoretical explanation.

Furthermore, the results of our work would apply to the regimes even when the TBG is not ferromagnetic. We did not just explain the two recent very high-profile experiments,

but also pointed out that our theory is applicable to many other moire systems such as strained twisted TMDs and monolayer graphene/hBN heterostructures. Our work will likely to inspire a large number of experimental and theoretical works for the study of current induced magnetization in moire systems.

Comment 5: “With that in mind, my assessment is that the calculation of the magnetization through the Berry curvature is interesting and timely, but not sufficiently original in my view to justify publication in Nature Communications. I would rather recommend publication in a more specialized journal.”

Reply 5: We thank the Referee for recognizing our work as interesting and timely. Through our reply, we hope the Referee can agree with us that:

1. The main claim of this work is not to use strain to enhance the valley polarization in TBG. Instead, we pointed out that a large magnetization can be generated by a small current due to the large Berry phase of the flat bands, combined with C_3 rotational symmetry breaking by strain. This current-induced magnetization is zero for conventional gapped bilayer graphene or TMDs, even with a large voltage bias.

In *strained* gapped bilayer graphene, we showed in Fig.R1 above that the current-induced magnetization is indeed three orders of magnitude smaller due to the small density of states.

2. The magnetoelectric effect we studied is the current-induced magnetization, and the magnetization direction is controlled by the direction of the current. In ferromagnetic systems with valley polarization and C_3 symmetry, a charged current cannot switch the direction of the magnetization of the sample.

3. Our results are general and it can be applied to multiple moire systems, including monolayer graphene on boron nitride substrate and twisted TMDs with strain, even when there is no spontaneous valley symmetry breaking in these systems. We believe that our work will attract a lot of experimental and theoretical effort to study the current-induced magnetization in moire systems.

4. Importantly, our work provides a solid understanding of the current-induced magnetization switching in two recent high-profile experiments.

5. Indeed, as pointed out by the Referee, our work is extremely timely. Young’s experimental work is only published in Science in December 2019, three months after our work is submitted to Nature Communications for review.

We believe that this work meets the standard of Nature Communications and we hope that the Referee can re-assess the importance of our work.

Changes made:

We thank the comments of the Referee which helped us to improve the presentation of the manuscript. In order to highlight the important results of our work, we have re-written the Abstract, part of the Introduction and part of the Discussion.

In particular, in both the Abstract and the Introduction, we emphasize that the current-induced magnetization can happen even when the TBG is not ferromagnetic (such as in the absence of valley polarization).

In the Discussion Section, we also added that “**and gapped bilayer graphene [59] with strain will exhibit similar magnetoelectric effects**, although the magnitude of the magnetoelectric response will depend on the details of the materials.” Here, Ref. 59 is the work by Nandkishore et al..

Referee 2:

Comment 1: In the manuscript ‘‘Giant Orbital Magnetoelectric effect and Current-induced Magnetization Switching in Twisted Bilayer Graphene’, the authors studied how the out-of-plane magnetic moment could switch under the in-plane electrical current. The authors constructed a model for the band structure in the presence of strain and the underlying substrate of BN. The authors found that the out-of-plane orbital magnetic moment of the Bloch states could become huge when the momentum is close to Gamma. Motivated by the recent experiments on magnetic moment switching by electrical current, the authors also calculated the magnetoelectrical response based on the linear response theory. The authors also proposed a phenomenological model for current switching.’’

Reply 1: We thank the Referee for the careful reading of our manuscript and for providing a precise and concise summary of the results of every section of our manuscript.

Comment 2: ‘‘I recommend its publication on nature communication as this article contains some impressive points:

1. The authors notice that the system is not in the quantum anomalous Hall regime when the current switching occurs. Although it is generally believed that the insulating state at commensurate filling could be understood from strong coupling limit, this observation partially justifies the authors' approach that treats the system as Fermi liquid. Furthermore, this model leads to predictions different from the theory based on edge modes.’’

2. This paper also contains some interesting points on symmetry analysis and reveals that the symmetry should be largely broken in order to explain the observed magnetoelectrical effects. This symmetry broken could result from the strain and aligned BN. The authors constructed a nice model for the strain effects.

Overall, it is a nice and interesting article. Therefore, I do recommend its publication on nature communication.’’

Reply 2: We thank the Referee for the careful reading of the manuscript and thoughtful comments on our work. We thank the Referee for his/her support of the publication of our manuscript.

References:

- [1] Edelstein, V. M. Spin polarization of conduction electrons induced by electric current in two-dimensional asymmetric electron systems. *Solid State Commun.* **73**, 233 (1990).
- [2] Ohta, T., Bostwick, A., Seyller, T., Horn, K., & Rotenberg, E. Controlling the electronic structure of bilayer graphene. *Science* **313**, 951 (2006).
- [3] Mak, K. F., Lui, C. H., Shan, J., & Heinz, T. F. Observation of an electric-field-induced band gap in bilayer graphene by infrared spectroscopy. *Phys. Rev. Lett.* **102**, 256405 (2009).

REVIEWERS' COMMENTS:

Reviewer #1 (Remarks to the Author):

After reviewing the manuscript and the response of the authors, I am persuaded that the work is novel and important enough to merit publication in Nature Communications.

The key finding of the paper is that a broken C3 symmetry in twisted graphene bilayers (due to strain) leads to a net orbital magnetization when a current is injected. The effect does not depend on time reversal symmetry being broken, and can happen at arbitrary filling factors, including in the metallic phase, and hence is different of the standard orbital magnetoelectric effect in Chern insulators and systems with anomalous Hall effects in general. This effect is greatly enhanced by the large density of states in the flat bands.

These results appear to explain the current switching effect observed recently by different groups and is of clear technological importance. The changes in the abstract and in the introduction have significantly improved clarity of the paper. Overall, this is a very interesting result in a very timely problem. For those reasons, I recommend publication in Nature Communications.

Reviewer #3 (Remarks to the Author):

This paper theoretically studies the giant orbital magnetic momentum induced by the small current due to the enhanced Berry curvature in twisted bilayer graphene with reduced symmetry due to strain and substrate. I have examined previous reviewer's comments and responses from authors, and find that author's rebuttal is reasonable. The giant orbital magnetization and consequent magnetization switching is a novel finding and offers a reasonable scenario for the experiment. Considering that the magnetization switching by current in terms of spin-orbit coupling usually requires huge current density, this mechanism without spin-orbit interaction is very promising also for the applications. Thus, I do recommend its publication in Nature Communications.